# Defining Antibody Seroprevalence and Duration of Humoral Responses to SARS-CoV-2 Infection and/or Vaccination in a Greek Community

**DOI:** 10.3390/ijerph19010407

**Published:** 2021-12-31

**Authors:** Ourania S. Kotsiou, Dimitrios Papagiannis, Evangelos C. Fradelos, Dimitra I. Siachpazidou, Garifallia Perlepe, Angeliki Miziou, Athanasios Kyritsis, George D. Vavougios, Georgios Kalantzis, Konstantinos I. Gourgoulianis

**Affiliations:** 1Faculty of Nursing, School of Health Sciences, University of Thessaly, Gaiopolis, 41110 Larissa, Greece; efradelos@uth.gr; 2Department of Respiratory Medicine, Faculty of Medicine, School of Health Sciences, University of Thessaly, Biopolis, 41110 Larissa, Greece; sidimi@windowslive.com (D.I.S.); perlepef19@gmail.com (G.P.); kellymiz95@yahoo.com (A.M.); thanoskyrit@hotmail.com (A.K.); dantevavougios@hotmail.com (G.D.V.); george.kalantzis4@gmail.com (G.K.); kgourg@med.uth.gr (K.I.G.); 3Public Health & Vaccines Lab, Department of Nursing, School of Health Sciences, University of Thessaly, Gaiopolis, 41110 Larissa, Greece; dpapajon@gmail.com

**Keywords:** antibody testing, Greece, semi-closed community, seroprevalence

## Abstract

Background: In this work, we aimed to evaluate antibody-response longevity to SARS-CoV-2 infection and/or vaccination in one of the Greek communities that was worst hit by the pandemic, Deskati, five months after a previous serosurveillance and nine months after the pandemic wave initiation (October 2020). Methods: The SARS-CoV-2 IgG II Quant method (Architect, Abbott, IL, USA) was used for antibody testing. Results: A total of 69 subjects, who previously tested positive or negative for COVID-19 antibodies, participated in the study. We found that 48% of participants turned positive due to vaccination. 27% of participants were both previously infected and vaccinated. However, all previously infected participants retained antibodies to the virus, irrespective of their vaccination status. The antibody titers were significantly higher in previously infected participants that had been vaccinated than those who were unvaccinated and in those that had been previously hospitalized for COVID-19 than those with mild disease. Conclusions: Antibody responses to SARS-CoV-2 infection were maintained nine months after the pandemic. Vaccination alone had generated an immune response in almost half of the population. Higher antibody titers were found in the case of vaccination in previously infected subjects and especially in those with severe disease leading to hospitalization.

## 1. Introduction

Humanity has faced an unequal fight with the SARS-CoV-2 pandemic for nearly two years. The duration of antibody responses after infection is unknown, and there is very little data beyond 35 days post-symptom onset [1].

Various intensities of decline in neutralizing antibodies after post-symptom onset have been reported [2,3,4,5,6]. While the evidence is increasing that disease severity is a factor that strongly correlates with the decay rate of neutralization. In that context, detectable levels of neutralizing antibodies against SARS-CoV-2 have been shown to start declining within three months of infection among mild and asymptomatic cases [4]. It is unlikely that this observation is a predictor of impermanent immunity and heightened risk of reinfection in the short term [2,3,4,5,6]. On the contrary, other data indicated that neutralizing antibody titers remain stable, ranging from 75 days to 6 months post-symptom onset in individuals with a broad spectrum of disease severity [2,3,5], even in patients with only mild-to-moderate COVID-19 [2].

At present, vaccines are our most potent tool to fight all the virus strains [6]. Similarly, much uncertainty exists about the persistence or decline of total antibodies following vaccination [6]. There is an urgent need for immunosurveillance studies to estimate the duration of post-vaccination immunity [6]. Effective and ethical response strategies to the COVID-19 pandemic can only be formulated once it is accurately determined if neutralizing antibodies are present and how long the post-infection and post-vaccination immunity will last [1,7].

We aimed to evaluate the humoral-response longevity to SARS-CoV-2 infection and/or vaccination in one of the communities in Greece that has been most affected by the pandemic, Deskati, after five months from a previous serosurveillance program (January 2021) and nine months after the initiation of the pandemic wave (October 2020).

## 2. Materials and Methods

A serosurveillance program was conducted in the municipality of Deskati on 6 June 2021 to investigate the duration of antibody responses to SARS-CoV-2 infection and/or vaccination five months after an initial serosurveillance study [8] and nine months after the initiation of the pandemic wave in the area (October 2020). All the residents of Deskati who had been recruited to the first serosurveillance program (conducted in January 2021 [9]) were invited to participate in the follow-up program by the local authority and had been notified of the time and place thereof. Participants were recruited by announcing the research in the media while local officials organized a one-month recruitment campaign. There were no exclusion criteria. The participants were analyzed to evaluate seroprevalence and antibody-response longevity to SARS-CoV-2 infection and/or vaccination.

This study was approved by the Ethics Committee of the University Hospital of Larissa, and all subjects provided written and oral informed consent. Following consent, demographic information and data regarding past PCR-confirmed COVID-19 infection and vaccination history were recorded on questionnaire forms for all participants. Following consent, demographics, somatometric characteristics, comorbidities, medication, and data regarding past PCR-confirmed COVID-19 infection that had been documented in the medical records, vaccination history, and previous SARS-CoV-2 antibody testing were recorded on questionnaire forms for all participants.

The SARS-CoV-2 IgG II Quant ELISA method (Architect, Abbott, IL, USA), a chemiluminescent microparticle immunoassay, was used for the qualitative and quantitative determination of IgG antibodies against the spike receptor-binding domain (RBD) of SARS-CoV-2 in serum specimens [8], with a sensitivity of 99.9% and specificity and 100% for detecting the IgG antibodies generated by prior infection or vaccination, as previously described [9,10].

Statistical analyses were performed with IBM SPSS Statistics for Windows, version 23.0, Armonk, NY: IBM Corp. The chi-square test and unpaired *t*-test were used to compare frequencies and parametric data between two groups, respectively.

## 3. Results

A total of 69 subjects with a mean age of 53.2 ± 12.5 years participated in the antibody surveillance program. Males were significantly older than females (Table 1). No difference in the other characteristics was detected. A total of 39.1% (*n* = 27) of the participants had a known past infection and 62.3% (*n* = 43) of the population had been vaccinated with at least one dose of a vaccine. There was no difference in comorbidities between people who had been vaccinated and unvaccinated people (55.8% vs. 39.1%, *p* = 0.151). Vaccinated adults were significantly older than those who had not been vaccinated (56.8 ± 11.6 vs. 46.4 ± 11.4, *p* = 0.001). There was no gender difference between the vaccinated and non-vaccinated participants. A total of 38.4% (10/26) of the non-vaccinated population and 39.5% (17/43) of the vaccinated population had been previously infected with SARS-CoV-2. Significantly higher antibody titers were detected in previously infected participants who had been hospitalized (8/27) than in previously infected participants with mild COVID-19 (19/27) (18,673 vs. 6627 AU/mL, *p* = 0.013).

SARS-CoV-2 seropositivity was 79.7% (*n* = 55) in the study population, achieved mainly by vaccine-acquired immunity (28/55 subjects, 51%), naturally-acquired immunity (12/55 subjects, 22%), or both (15/55 subjects, 27%). The mean antibody title was 7202 ± 1222 AU/mL. A total of 56% of the participants who had received a COVID-19 vaccination were partly or fully vaccinated with the Oxford–AstraZeneca vaccine, 42% received at least one dose of the Pfizer–BioNTech vaccine, and 2% received the single-shot Johnson & Johnson vaccine.

A total of 48% of the whole population switched to testing positive due to vaccination, while 46% of the whole population retained positive antibodies due to natural (47%) and/or acquired antibody production (53%) 30 ± 29 days after the second dose of a vaccine (Figure 1). Only 6% of the population remained antibody negative, i.e., not infected or vaccinated (Figure 1). None of the previously-positive tested subjects turned to a negative result in the antibody testing.

Antibody titers were significantly higher in previously infected and vaccinated individuals than previously infected, unvaccinated participants (8132 ± 6501 vs. 3615 ± 2203, *p* = 0.027). No difference was detected in antibody titers between uninfected vaccinated subjects and those previously infected but unvaccinated. Moreover, no age difference in antibody titers was detected among only vaccinated, only infected, or both vaccinated and infected populations. All previously infected participants tested positive for antibodies nine months after the initiation of the pandemic wave in the area (October 2020). Figure 2 presents the methodology and the main results of this study.

## 4. Discussion

In the present study, we evaluated the duration of antibody responses to SARS-CoV-2 infection and/or vaccination in one of the communities in Greece that has been most affected by the pandemic, Deskati, in a follow-up program five months after the first serosurveillance program conducted in January 2021 and almost nine months after the initiation of the pandemic wave (October 2020). We found that SARS-CoV-2 seropositivity was 79.7%, achieved mainly with vaccination. Vaccinated adults were significantly older than those who had not been vaccinated. A total of 62.3% of the population had been vaccinated with at least one dose of a vaccine. One third (15/55) of the vaccinated population had a past infection. Regardless of the vaccination status of the individuals, all previously infected participants retained antibodies to the virus nine months after the infection. The antibody titers were significantly higher in participants that had a severe, rather than a mild, case of COVID-19. The antibody titers were significantly higher in previously infected, vaccinated participants than previously infected, unvaccinated participants. Moreover, we found that 48% of the population previously tested for COVID-19 antibodies switched to testing positive due to vaccination, and 46% of them retained positive antibodies due to natural or acquired antibody production.

Sensitivity and specificity thresholds of serological assays are critical for epidemiological considerations in unique environments, and there should be a balance between them [8], as the epidemiological implications of disproportionate false negatives or false positives are profound [6]. Accordingly, the ELISA method used has very high sensitivity and specificity for detecting the IgG antibody [8,10].

One of the most widely discussed epidemiological concepts surrounding COVID-19 is the possibility of achieving herd immunity [6]. Herd immunity can either be natural herd immunity or achieved by controlled vaccination programs [11]. Recent seroprevalence data show that no country has reached herd immunity through natural acquisition. In countries where no official lockdown measures were enforced throughout the pandemic, there were only low seroprevalence levels [12], usually much lower than 10%. A recent meta-analysis found that SARS-CoV-2 seroprevalence in the general population varied from 0.37% to 22.1%, with a pooled estimate of 3.38% (95%CI 3.05–3.72%; 15 879/399 265). On a regional level, seroprevalence varied from 1.45% (0.95–1.94%, South America) to 5.27% (3.97–6.57%, Northern Europe) [13]. We suggest that herd immunity thresholds could have been reached in Deskati, as SARS-CoV-2 seropositivity was 79.7% in the study population, attributed mainly to vaccine-acquired immunity and, in approximately one fourth of the population, to natural infection. However, larger studies are needed and are currently being carried out to confirm this suggestion.

A recent study observed that seroprevalence of SARS-CoV-2 antibodies in over 6000 healthcare workers in Spain was higher in moderate-to-severe disease (median antibody titer: 13.7 AU/mL) compared with mild (6.4 AU/mL) and asymptomatic (5.1 AU/mL) infection [14]. Similarly, among previously infected seropositive individuals, we found significantly higher antibody titers in those who had been hospitalized than in non-hospitalized participants with mild COVID-19 (18.6 AU/mL vs. 6.6 AU/mL, *p* = 0.013). Interestingly, Montenegro et al. showed that almost 40% of the mild symptomatic cases, that were followed up by general practitioners during the peak months of the pandemic, were seropositive [15].

Varona et al. [14] also supported that antibody titers were higher in older people (>60 years: 11.8 AU/mL) compared with younger people (<30 years: 4.2 AU/mL). On the contrary, our study detected no age differences in antibody titers among vaccinated, infected, or both vaccinated and infected populations.

A total of 62.3% of the population had been vaccinated with at least one dose of a vaccine. Vaccinated adults were significantly older than those who had not been vaccinated, which was an expected finding as people who are elderly are the most vulnerable to COVID-19 [6]. One third (15/55) of the vaccinated population had a past infection. Research demonstrates that vaccinating post-infected individuals substantially enhances their immune response and confers strong resistance against variants of concern, including the B.1.617.2 (Delta) variant. More specifically, it has been documented that serum spike IgG Ab levels were higher after vaccination in infected patients than after natural infection 180 days after the completion of the second vaccine dose or natural infection [16,17]. It is widely accepted that individuals who were both previously infected with SARS-CoV-2 and given a single dose of the vaccine gained additional protection against SARS-CoV-2 [16]. This study confirms that notion showing that the antibody titers were more than doubled in vaccinated compared with unvaccinated, previously infected participants.

We found no difference in antibody titers between vaccinated, uninfected subjects and unvaccinated, previously infected subjects. To date, there has been little agreement on the long-term protection conferred by a previous infection compared to vaccination. On the one hand, there is data supporting that natural immunity confers longer-lasting and stronger protection against infection, symptomatic disease, and hospitalization caused by the Delta variant of SARS-CoV-2, compared to the BNT162b2 two-dose, vaccine-induced immunity [6,16]. On the other hand, other studies demonstrated that vaccination alone led to higher antibody titers compared to natural infection [6,17].

Vaccination changed the landscape in Deskati, as 48% of the participants switched to testing positive due to vaccination, while 46% of the participants retained positive antibodies due to natural or acquired antibody production. We documented that regardless of the vaccination status of an individual, all previously infected participants retained antibodies to the virus nine months after the pandemic wave (October 2020). Many studies measured a decline of IgG antibodies several weeks post symptom onset [18,19]. As previously mentioned, a debate persists regarding not only the intensity but also the duration of the IgG antibody responses to SARS-CoV-2. A previous study demonstrated that 40% of asymptomatic and 12.9% of symptomatic individuals became seronegative during a study period of 8 weeks [20]. However, in accordance with the present results, most observations showed that IgG levels against SARS-CoV-2 remained relatively stable within a three-to-five-month observation period after symptom onset [2,3,5,6,19,21,22,23,24,25,26,27]. Regular re-exposure to the virus may help sustain higher neutralizing antibody levels by stimulating memory B cells to mount a rapid and effective humoral response [6]. Varona et al., found that seropositive healthcare workers of the Spanish Hospitals Group remained seropositive after 9 months but with a significant decline in antibody titers and two distinct antibody dynamic profiles were identified (declining vs. stable). Independent factors associated with longer persistence of antibodies were symptomatic infection and higher exposure to COVID-19 patients [24,28,29], age, BMI, and immunosuppression [30].

It has been shown that titers of IgM and IgG antibodies against the receptor-binding domain (RBD) of the spike protein of SARS-CoV-2 decrease significantly over this time period, with IgA being less affected. Memory B cells display clonal turnover after 6.2 months in a manner consistent with antigen persistence, and the antibodies that they express have increased potency, indicative of continued evolution of the humoral response [30]. Dan et al. supported that spike-specific memory B cells were more abundant at 6 months than at 1 month after symptom onset [25]. Another study documented that memory B lymphocytes persisted and displayed functional hallmarks of antiviral immunity by expressing receptors capable of neutralizing the virus when expressed as monoclonal antibodies for at least three months [31]. On the other hand, delayed or absent type I and III interferons (IFN-I and IFN-III) and early or late response, together with robust cytokine and chemokine production, are implicated in the development of severe COVID-19 [32].

Although all individuals who participated in the present study were respondents, a limitation of this study was that they constituted only a small fraction of the population in Deskati. Moreover, we cannot exclude those participants who might have characteristics associated with willingness to participate. The population was limited to one geographic area, which translated into a lack of generalizability. A strength of this study was that individuals had been enrolled at a late phase of the second epidemic wave and the seroprevalence data could paint a clear picture of the immunoprotection in the targeted population.

## 5. Conclusions

Antibody responses to SARS-CoV-2 infection were maintained nine months after the pandemic in the studied population. Higher antibody titers were found in the case of vaccination in previously infected subjects and subjects that were previously hospitalized for COVID-19 than those with mild disease.

Effective and ethical response strategies to the COVID-19 pandemic can only be formulated once the antibody seroprevalence and duration of humoral responses to SARS-CoV-2 infection and/or vaccination at the population level have been accurately determined. Understanding the temporal profile, by which circulating antibody classes are ranked following SARS-CoV-2 infection and vaccination, is essential for interpreting immunity dynamics and implementing public health measures that rely heavily on up-to-date knowledge of transmission dynamics.

This study has shed light on the humoral-response longevity to SARS-CoV-2 infection and/or vaccination in one of the communities in Greece that has been most affected by the pandemic. Higher antibody titers were found in the case of vaccination in previously infected subjects. Memory responses are responsible for protection from reinfection and are essential for effective vaccination. The observation that memory B cell responses did not decay after 9 months, but instead continued to evolve, strongly suggests that individuals who are infected with SARS-CoV-2 could mount a rapid and effective response to the virus upon re-exposure. This study highlighted the effectiveness of epidemiological studies to continuously evaluate the baseline amount of disease occurrence and seroprevalence due to naturally or vaccine-acquired antibody production in a community.

## Figures and Tables

**Figure 1 ijerph-19-00407-f001:**
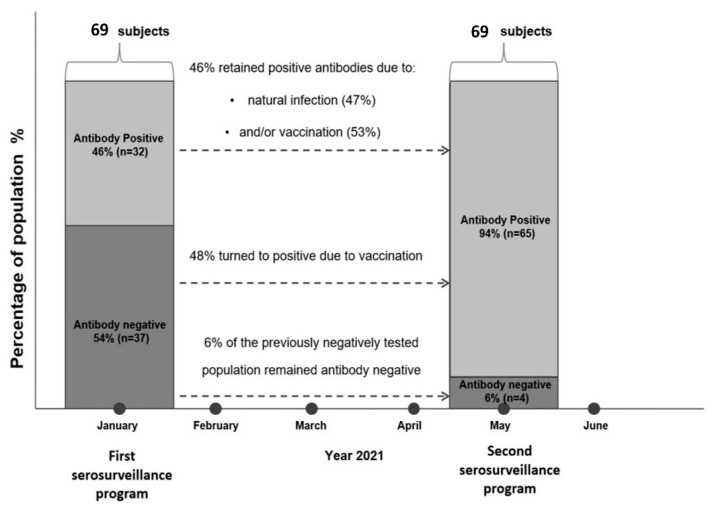
Antibody responses to SARS-CoV-2 infection and/or vaccination in Deskati, five months after the first serosurveillance and nine months after the initiation of the pandemic wave (October 2020).

**Figure 2 ijerph-19-00407-f002:**
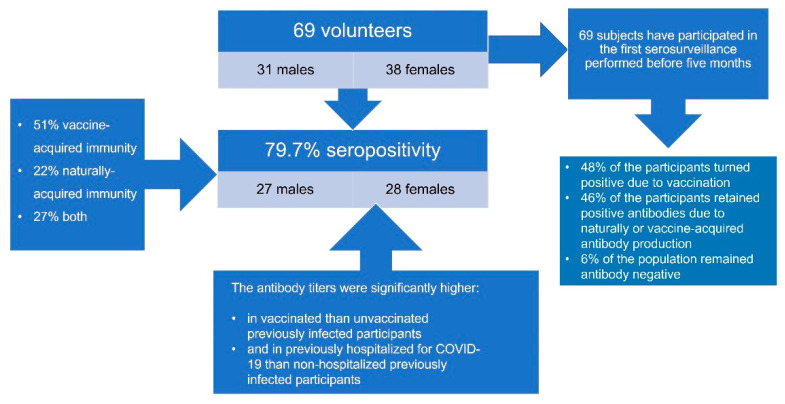
Analytic framework illustrating the methodology and main results of this study.

**Table 1 ijerph-19-00407-t001:** Characteristics of the study population, stratified by gender (N = 69).

Variable	Total (*n* = 69)	Males (*n* = 31)	Females (*n* = 38)	*p*-Value
Age (years)	53.2 ± 12.5	58.0 ± 13.2	49.3 ± 10.6	0.005 ^#^
BMI (mg/kg^2^)	25.0 ± 10.0	22.0 ± 12.0	27.0 ± 7.0	0.093 ^#^
Comorbidities, *n* (%)	33 (47.8)	17 (54.8)	16 (42.1)	0.311 *
On medication, *n* (%)	32 (46.4)	17 (54.8)	15 (39.5)	0.397 *
Previous infection confirmed, *n* (%)	27 (39.1)	15 (48.4)	12 (31.6)	0.181 *
Vaccinated, *n* (%)	43 (62.3)	22 (71.0)	21 (55.3)	0.251 *
Seropositive, *n* (%)	55 (79.7)	27 (87.9)	28 (73.7)	0.541 *
Antibody titers (AU/mL)	7202 ± 1222	8084 ± 1266	6373 ± 1192	0.580 ^#^

Note: Data are expressed as mean ± SD or as frequencies (percentages). ^#^
*t*-test; * *chi-square*.

## Data Availability

The data that support the findings of this study are available on request from the corresponding author, O.S.K.

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
