# Peer review of "Defining Antibody Seroprevalence and Duration of Humoral Responses to SARS-CoV-2 Infection and/or Vaccination in a Greek Community"

_ijerph, 2021, doi:10.3390/ijerph19010407_

Round 1

Reviewer 1 Report

In this manuscript, Ourania et al analyzed antibody response longevity to SARS-CoV-2 infection and/or vaccination in Deskati population, which include 69 subjects. Their results showed a 79.7% seropositivity in the study population, which include vaccine-acquired immunity(51%), naturally-acquired immunity(22%) and both(27%), with mean antibody titer at about 7202AU/ml. They found antibody responses to SARS-CoV-2 infection could be maintained 9 months after pandemic. There’s no difference in antibody titer between uninfected vaccinated participants and previously infected without vaccination participants. Interestingly, individuals previously being infected had higher antibody titers after vaccination. This study is very important to understand humoral response longevity to SARS-CoV-2 infection and/or vaccination and make effective strategies to control COVID-19 pandemic.

Specific comment:

  1. Some parts of this manuscript could easily make readers confused. For example, in third paragraph of the results section (line 98), the author said ‘48% of the previously negative tested participants switched to testing positive due to vaccination’. What does 48% mean? 48% of negative participants or 48% of total population in this study? From the sentence, it looks like 48% of previously negative individuals switched positive. But from what shown in Figure 1, it looks like 48% means the percentage from total population studied in this manuscript. The figure is also not very clear to explain what the number mean. Data is the most important part of this study, so the authors should pay more attention to express it clearly in sentences and figures.
  2. Since this study compared the antibody titers between SARS-CoV-2 infection and vaccination, it would be important and helpful if the author could provide information about the vaccine received by subjects in this study, such as the manufacturer.

Author Response

RESPONSE TO REVIEWER 1

  1. In this manuscript, Ourania et al analyzed antibody response longevity to SARS-CoV-2 infection and/or vaccination in Deskati population, which include 69 subjects. Their results showed a 79.7% seropositivity in the study population, which include vaccine-acquired immunity( 51%), naturally-acquired immunity(22%) and both(27%), with mean antibody titer at about 7202AU/ml. They found antibody responses to SARS-CoV-2 infection could be maintained 9 months after pandemic. There’s no difference in antibody titer between uninfected vaccinated participants and previously infected without vaccination participants. Interestingly, individuals previously being infected had higher antibody titers after vaccination. This study is very important to understand humoral response longevity to SARS-CoV-2 infection and/or vaccination and make effective strategies to control COVID-19 pandemic.

RESPONSE: We are delighted to receive positive feedback from you. We really thank you for taking the time and energy to help us improve this paper. We have carefully studied the comments and suggestions and revised our paper accordingly.

  1. Specific comment: Some parts of this manuscript could easily make readers confused. For example, in third paragraph of the results section (line 98), the author said ‘48% of the previously negative tested participants switched to testing positive due to vaccination’. What does 48% mean? 48% of negative participants or 48% of total population in this study? From the sentence, it looks like 48% of previously negative individuals switched positive. But from what shown in Figure 1, it looks like 48% means the percentage from total population studied in this manuscript. The figure is also not very clear to explain what the number mean. Data is the most important part of this study, so the authors should pay more attention to express it clearly in sentences and figures.

RESPONSE: Thank you for this valuable remark. In the revision, we have defined this issue (page 1, line 19; page 3, lines 110-115; page 4, lines 145-148; and page 5 , lines 205-207). We apologize for the confusion.

  1. Since this study compared the antibody titers between SARS-CoV-2 infection and vaccination, it would be important and helpful if the author could provide information about the vaccine received by subjects in this study, such as the manufacturer.

RESPONSE: Thank you for this valuable remark. In the revision, we have provided information about the vaccine (page 3, lines 104-109).

We greatly appreciate your detailed and constructive comments that helped improve the manuscript. We trust that all your comments have been addressed accordingly in a revised manuscript. We hope you find these revisions rise to your expectations.

Reviewer 2 Report

This is an interesting paper evaluating Seroprevalence and duration of SARS-CoV-2 humoral response in a brief cohort of Greek Community.

The methodology is correct, and the presentation of the results and the discussion is also appropriate.

I would highlight some points to improve:

  • We recommend use a graph to better understanding the metodology.
  • A clearer comparation between vaccinated and non-vaccinated population is necessary in results
  • They must update bibliographic search in two key points:
    1. On seroprevalence. Some large cohorts are missed in the references, especially in the Mediterranean area (example: DOI 10.1093 / ije / dyaa277).
    2. With respect duration of anti-SARS-cov-2 humoral response, and this point is even more important to improve the search on previous published data- Thus, some recent and important series are missed, again especially in the Mediterranean area (example: kinetics of anti-SARS-CoV-2 antibodies over time. Results of 10 month follow up in over 300 seropositive Health Care Workers; DOI: 1016/j.ejim.2021.05.028). This should be considered in the manuscript and include these newest studies, also commenting on them in the discussion.
  • English needs to be edited

Author Response

RESPONSE TO REVIEWER 2

  1. This is an interesting paper evaluating Seroprevalence and duration of SARS-CoV-2 humoral response in a brief cohort of Greek Community. The methodology is correct, and the presentation of the results and the discussion is also appropriate.

RESPONSE: We are delighted to receive positive feedback from you. We really thank you for taking the time and energy to help us improve this paper. We have carefully studied the comments and suggestions and revised our paper accordingly.

  1. I would highlight some points to improve. We recommend use a graph to better understanding the methodology.

RESPONSE: Thank you for this comment. In the revision, we have added a graph to explain the methodology and results of this study better (Figure 2).

  1. A clearer comparation between vaccinated and non-vaccinated population is necessary in results

RESPONSE: Thank you for this suggestion. In the revision, we have provided a clearer comparison between vaccinated and non-vaccinated population (page 2-3, lines 90-99).

  1. They must update bibliographic search in two key points: On seroprevalence. Some large cohorts are missed in the references, especially in the Mediterranean area (example: DOI 10.1093 / ije / dyaa277). With respect duration of anti-SARS-cov-2 humoral response, and this point is even more important to improve the search on previous published data- Thus, some recent and important series are missed, again especially in the Mediterranean area (example: kinetics of anti-SARS-CoV-2 antibodies over time. Results of 10 month follow up in over 300 seropositive Health Care Workers; DOI: 1016/j.ejim.2021.05.028). This should be considered in the manuscript and include these newest studies, also commenting on them in the discussion.

RESPONSE: Thank you for this valuable remark which helped us to improve the quality of the article. In the revision, we have updated bibliographic search as suggested, including newest studies and commenting on them.

  1. English needs to be edited

RESPONSE: Thank you for this comment. We regret there were problems with the English. The paper has been carefully revised by a native English speaker to improve the grammar and readability.

We found your feedback very constructive. We tried to be responsive to your concerns. We really thank you for taking the time and energy to help us improve this paper.

Round 2

Reviewer 2 Report

The authors have made a meritorious effort to respond to the requirements to improve the manuscript.
We currently believe that it adds value in terms of seroprevalence in COVID-19